# Visceral Leishmaniasis and the Skin: Dermal Parasite Transmission to Sand Flies

**DOI:** 10.3390/pathogens11060610

**Published:** 2022-05-24

**Authors:** Sahaana Arumugam, Breanna M. Scorza, Christine Petersen

**Affiliations:** 1Department of Epidemiology, College of Public Health, University of Iowa, Iowa City, IA 52242, USA; sahaana-arumugam@uiowa.edu (S.A.); breanna-scorza@uiowa.edu (B.M.S.); 2Immunology Program, Carver College of Medicine, University of Iowa, Iowa City, IA 52242, USA; 3Center for Emerging Infectious Diseases, University of Iowa Research Park, Coralville, IA 52241, USA

**Keywords:** parasite burden, transmissibility, visceral leishmaniasis, infectiousness, dermal immune environment

## Abstract

Visceral leishmaniasis is a parasitic disease with significant dermal tropism. The skin is an important site of infection contributing to parasite transmission to naïve sand flies, but understanding how parasitism of host skin and the related immune microenvironment supports or prevents skin parasite replication is now the focus of major investigation in the field of leishmaniasis research. Here, we review dermatoimmunology during visceral leishmaniasis (VL), dermal *Leishmania* parasite burden, and the role of skin parasitism in transmissibility to sand fly vectors. First, we discuss the epidemiology of VL amongst dogs, the primary zoonotic reservoir for human infection. We explore the association between spatial distribution and the burden of parasites in the skin in driving outward transmission. Factors associated with parasite persistence in the skin are examined. We discuss systemic immunity during VL and what is known about immunological correlates in the skin microenvironment. Finally, we touch on factors egested into the skin during *Leishmania* inoculation by sand flies. Throughout, we discuss factors associated with the early and chronic establishment of *Leishmania* parasites in the skin and the role of the dermal immune response.

## 1. Introduction

Visceral leishmaniasis (VL) is caused by *Leishmania donovani complex* trypanosomatid protozoan parasites. It is a fatal disease in humans with >90% case fatality within two years if left undetected and untreated [1]. VL due to *L. donovani* spp. is a zoonotic disease that causes 50,000 to 90,000 new human cases, more than 50,000 deaths, and countless canine cases per year [2,3]. This review seeks to explore recent evidence regarding the dermotropic nature of visceralizing species of *Leishmania*, the dermal immune environment during VL, and the role of dermal parasite burden in outward transmission to naïve sand flies.

We utilized the PubMed database to compile data from primary manuscripts and review articles regarding visceral leishmaniasis species and the skin occurring over the last two decades. To our knowledge, this is the first review of this subject. This literature review was carried out using the PubMed database. Key search words used included “leishmaniasis”, “xenodiagnoses”, “infectivity”, “transmission”, “immunology”, “skin”, and “dermal”. Original articles were prioritized over case reports or conference proceedings, and articles were initially chosen from the last 20 years except where seminal papers from prior years were used or suggested by reviewers.

As new evidence emerges regarding how skin parasite burden predicts infectiousness from host to vector, we believed it was necessary to delve further into the literature on the implications of the dermal immune environment’s role in skin parasite burden and subsequent outward transmission. It is important to better understand the factors affecting outward transmission from infected hosts, as this is a key component of the transmission cycle for preventive interventions to be targeted. Elucidation of the immunological mechanisms that exacerbate dermal parasite accumulation may have implications for the skin’s potential role in outward transmission. This could also mean the skin is a prime target for immunological interventions, such as topical biologics, in breaking the parasite transmission cycle.

The review details the epidemiology of VL and the importance of canid reservoirs in transmission. We then go on to discuss how both the burden and spatial distribution of dermal parasite burden are related to outward transmission to vectors. We then examine the causes of parasite persistence in bite sites and the contribution parasite persistence has to the buildup of parasite burden in the skin. We first look at the early and then the chronic establishment of dermal parasite burden from an immunological perspective. Lastly, we consider the immune influence of components transferred from sand flies into the skin during bites and the subsequent inflammation from a sand fly bite in the skin. Our goal is to review the literature regarding multiple factors that contribute to dermal parasite burden and emphasize its role in outward transmission.

## 2. Epidemiology

*Leishmania infantum* is the primary causative strain of VL across the Mediterranean basin (Southern Europe, North Africa, the Middle East) and the Americas [4]. *Leishmania donovani* is the primary cause of VL in East Africa and South Asia [5]. As of 2020, the bulk of VL cases occurred amongst impoverished, vulnerable communities, with 90% of cases occurring in just 10 countries: India, Sudan, South Sudan, Ethiopia, China, Eritrea, Kenya, Somalia, Yemen, and Brazil [6]. *L. donovani complex* spp. are also known to occasionally cause cutaneous leishmaniasis (CL) in Sri Lanka. *L. infantum* is known to cause CL in northern India, northern Pakistan, northern Iran, and Turkey. [7,8]. *L. donovani* is currently considered anthroponotic, although a role for animal reservoir hosts is possible [9,10], and *L. infantum* resides in canine reservoirs in the Americas as well as the Mediterranean basin. *L. infantum* was imported to North America via hunting dogs from Southern Europe [11], and *L. infantum* within the North American hunting hound population is enzootic [12,13,14]. Using data collected over 15 years, Toepp et al. calculated an incidence rate of 25 cases of VL per 1000 dogs [15]. Regions of the Mediterranean such as Italy have reported CanL incidence rates as high as 9.5% [16]. Surveillance studies of dogs in endemic areas of Brazil put prevalence as high as 22% by PCR testing [17]. Parasites from vertically infected North American hounds are still infectious to naïve competent sand flies in an experimental setting [18]. Three species of *Leishmania* competent sand fly vectors are thought to be present in North America: *Lutzomyia anthophora*, *Lu. diabolica*, and *Lu. shannoni* [19,20]. Sand fly vector distribution patterns in Europe have changed in the past few decades primarily due to climate change and other environmental factors. Changes in vector distribution patterns in North America may similarly occur due to environmental changes [21,22]. Although *Lu. shannoni* in North America has not been proven to be a leishmaniasis vector according to the modified Killick-Kendrick criteria for incriminating natural vectors [23], it has been shown to possibly be a vector for *L. infantum* in Brazil [24], and changes in vector distribution may potentially allow North American sand fly species to fulfill the vector incrimination criteria. This could potentially pose a zoonotic risk of outward domestic transmission from naturally infected canine reservoirs to proximate human populations, particularly in the Southwest, South, and Southeastern U.S., where sand flies are endemic [19,20]. Canine reservoir status and similarities between human and canine *L. infantum*-specific immunology make dogs a public health-relevant animal model for studying VL pathogenesis, intervention, and therapeutic strategies.

## 3. Transmission and Dermal Parasite Burden

*Leishmania* parasites are transmitted between mammalian hosts via female phlebotomine sand flies and vertical, transplacental infection [25,26,27]. Although VL is primarily known to affect organs such as the bone marrow, liver, and spleen, there is evidence to suggest parasitism of the skin in most infected mammals. Dermal parasite burden may accumulate due to the skin being the initial inoculation site and maintained through multiple immune mechanisms, including a unique inflammatory environment in the skin, regulatory cell proliferation, protein induction, macrophage recruitment, T cell exhaustion, and parasite immune evasion factors. The dermotropic nature of *Leishmania* has been shown to be critical for outward transmission to sand flies, with skin parasite burden being a predominating factor for infectiousness [18,28].

*L. infantum* has been shown to be significantly dermotropic, and skin parasite burden is highly correlated with parasite transmission to sand flies, even in congenitally infected dogs [18,28]. Skin parasite burden correlates with parasitemia, with skin parasite burden consistently higher than blood parasite burden [18]. Parasitemia, therefore, can be comparatively lower and is not always the best predictor of outward transmission [18,29]. Splenic parasite burden, generally thought to be one of the most parasitized tissues in VL, was comparable to skin parasite burden [18,28]. One study in dogs reported significant positive correlations between parasite load in skin and sand fly infection rate and sand fly parasite load post xenodiagnoses [30].

Although there was a correlation between the dermal parasite burden of dogs and transmission to sand flies, it is not clear if there is a definitive correlation between disease severity and transmission [28]. Laurenti et al. showed that the clinical severity of dogs infected with *L. infantum* was inversely correlated with the infection rate of sand flies in a xenodiagnosis study; all but one asymptomatic dog was able to transmit parasites to a naïve vector while a lower proportion of symptomatic dogs was able to do so [31]. Another study found that the infection rate of naïve sand flies fed on infected dogs had no correlation with the symptoms or clinical group of the dog [32]. Despite these findings, several other studies have found significant positive correlations between clinical severity and infectiousness. Courtenay et al. found that seropositive dogs in Brazil undergoing xenodiagnosis exhibited a positive correlation between clinical disease and infectiousness, and only 17% of the most highly infectious dogs accounted for >80% of sand fly infections [33]. A similar positive correlation between the clinical severity of dogs and infectiousness to sand flies experimentally has been reported in several other xenodiagnoses studies in Brazil [34,35,36,37,38] and one study in Colombia [39]. A study in the U.S. showed that dogs with mild or moderate VL had the highest transmissibility from the skin to sand fly vector, while severe disease did not [18]. Another xenodiagnoses study investigating ear skin, specifically from Brazilian dogs, found that relative numbers of *L. infantum* in ear skin increased with the duration and severity of infection [28]. A meta-analysis of data published up until 2009 found that the proportion of infectious dogs is significantly positively correlated with clinical severity [40]. Xenodiagnoses studies in other mammalian models such as hamsters showed that *L. donovani* transmission from sick hamsters to sand flies was surprisingly low, but new flies fed on the same site acquired significantly more infections [41]. In *L. major* infected BALB/c mice, repeated sand fly bites increased the parasite loads in the skin but did not alter infectiousness [42]. Taken together, infectiousness to sand flies was most associated with high skin parasite burden, while disease state and parasitemia were not as predictive of infectiousness.

Experiments investigating outward transmission from human skin to naïve sand flies, emphasizing the role of skin in transmissibility, have also been conducted. In humans, xenodiagnoses studies in VL patients due to *L. donovani* in Bihar, India, have shown that 55% of patients with VL were infectious to sand flies before treatment. Modeling using this data demonstrated that every 10-fold increase in skin qPCR parasite load increased the odds of infecting a naïve sand fly 1.0–5.53 times [43]. A study from Ethiopia used skin microbiopsies to detect *L. donovani* parasites. The study reported that microbiopsies had a 10-fold smaller volume of blood than finger stick samples yet yielded higher *Leishmania* DNA rates, indicating the skin may be a more accessible source of parasites for diagnostics compared with blood [44]. Volunteers with a history of VL were also just as likely as healthy volunteers to test positive for *L. donovani* on microbiopsy, meaning there are many more asymptomatic cases than previously thought [44]. In each endemic location, even small numbers of infectious dogs or humans have the capacity to successfully infect a proportionate amount of sand flies with parasites. While it is known that *L. donovani* parasites persist in human skin, it remains to be definitively confirmed whether infected human skin itself promotes *L. donovani* transmission to sand flies. Xenodiagnoses studies of HIV patients coinfected with *L. infantum* have been performed, and these studies found that a significant proportion of coinfected patients were highly infectious to sand flies [45,46]. One xenodiagnoses study conducted in Spain found two immunocompetent VL patients with the active but untreated disease were able to transmit *L. infantum* to sand flies [47]. The studies discussed above are part of a growing body of evidence highlighting the potential infectious capability of VL-infected human skin to sand fly vectors in an experimental setting.

Post-kala-azar dermal leishmaniasis (PKDL) is a serious cutaneous manifestation of VL that can occur after VL drug treatment presenting as nodular/papular or macular lesions [48]. Risk factors are believed to be immunosuppression or an immune reaction after VL drug treatment [48]. Patients with PKDL can infect naïve sand flies, illustrating a further example of outward transmission from the skin [49,50]. Over 50% of PKDL patients were capable of transmitting parasites to naïve sand flies. Of the PKDL patients that did demonstrate outward transmission, a higher skin parasite load by a factor of 10 was observed when compared with PDKL patients that were incapable of outward transmission [51]. Further studies investigating nonlesional skin of both symptomatic and asymptomatic VL patients are needed to further elucidate the potential of infected human skin for outward transmission and whether exposed areas are more likely to transmit parasites due to particular aspects of skin parasite distribution.

## 4. Dermal Distribution of *Leishmania* and Outward Transmission

*L. donovani* spp. have been shown to distribute heterogeneously in the skin of RAG-deficient mice lacking T and B cells after infection, creating heterogeneous pools of parasites in the skin that correlate with infectiousness and transmissibility [29,52]. The larger and denser a parasite pool is, the more likely transmission from that animal to a sand fly. The patchy distribution of parasites in the skin is an important finding because although patchiness decreases the expected number of sand flies acquiring parasites, it increases the infection load of a sand fly when it does find a particular patch, increasing transmissibility. Reservoir animals such as dogs can be “super-spreaders”, in which a small percentage of dogs harbored almost 90% of total parasites [28]. If a small percentage of reservoirs not only harbor the majority of parasites but distribute them in the skin in such a heterogenous way as to increase transmissibility, those dogs may be especially infectious to sand flies, posing an increased threat to human and animal health in endemic locations. Multiple blood feedings can also increase the frequency of parasite transmission [53]. The patchiness of *Leishmania* parasite burden in the skin has been shown to be true in an immunosuppressed (RAG1 deficient) mouse model, but it has not yet been shown to be true of *L. infantum* in a canine model. We speculate that based on the dermotropic nature of *L. infantum* and patchiness in a mouse model, the skin of infected dogs would also have heterogeneous, patchy distribution of parasites. This skin parasite landscape will be important for prediction models of infectiousness of other important hosts, dogs, or people. Modeling skin parasite residence is significant when sampling skin because it might explain confounding factors that affect host infectiousness and outward transmission. Doehl et al. used an endogenously fluorescent parasite model intravenously injected into a RAG-deficient mouse and combined data with spatial point pattern analysis to identify several model mechanisms to determine why the skin parasite landscape was patchy [54]. Their model suggested that parasites were initially seeded into the skin before uptake by myeloid cells [54]. Parasitized myeloid cells then aggregated in patches similar to granulomas. Uninfected myeloid cells recruited to cellular aggregates were infected by parasites from the initial seeding patch, creating self-propagating clusters of patches. Parasite patch size was highly variable, but patch centers tended to be more densely parasitized than the periphery, increasing the likelihood of parasite uptake from the center of patches [54]. Higher parasite loads accompanied by a more intense inflammatory infiltrate have been reported in the ear, ungual, and ventral abdomen regions of infected dogs [36], but other studies report no preferred skin site of parasite distribution on a micro level [54]. Skin wounds and trauma have been shown to recruit monocytes to the deep dermal layer as early as neutrophils infiltration [55]. Patients with Hansen’s disease (leprosy) caused by *Mycobacterium leprae* experience severe dermal lesions and inflammation due to infection, with numerous manifestations of dermal pathology. Hansen’s disease-associated dermal reactions can be misdiagnosed as PKDL, as they are similar to a dermal lesion [56]. It may be that the inflammatory environment in the skin during comorbidities that create lesional skin predisposes the skin to both infection via vector and parasite patch development due to the overwhelming presence of macrophages at lesional sites in the skin prior to *Leishmania* infection. We speculate that the initial bite serves as skin trauma that not only recruits monocytes but also sets up a future patch location. This is especially important for animals that live in groups, such as hunting dogs, whose faces and ears are often traumatized by other dogs, setting up areas of high monocyte infiltration that can then become an initial seeding parasite patch that self-propagates via clustering during chronic infection. Whether parasites prefer proximity to vasculature for uptake of nutrients is unclear, skin parasite distribution in relation to vasculature needs to be studied more closely. The spatial distribution and density of parasite distribution in the skin of infected mammals is emerging as a critical factor in outward transmission to sand flies.

## 5. Parasite Persistence at Bite Sites

The sand fly is a telmophage; its proboscis lacerates the skin of the host, causing localized damage in the dermis and creating a blood pool on which to feed. This allows for parasite uptake from both dermal blood vessels and dermal macrophages with phagocytosed parasites [29]. Similar to patches that occur in areas of trauma, there has been a specific study of parasite persistence in the skin at bite sites. Parasite persistence is as important to consider as spatial organization of parasites for outward transmission and belie a temporal component to parasite accumulation and density of parasitic patches in the skin. After the early stages of infection, parasites persist in patches for months to years, while chronic infection develops with consequent immune modulation of parasite-controlling mechanisms. *L. infantum* parasites persisted for at least six months in ulcerative skin lesions at primary bite sites, but parasites were also observed to persist for six months in normal-looking skin at a secondary bite site [57]. Sand flies were also found to acquire parasites after feeding on lesions at the primary bite site [57]. These findings allude to the ability of parasitized yet noninflamed or lesional skin to continue to be infectious to sand flies. Parasites can build up in the skin over time, even in clinically healthy mammals, and those parasites can still be outwardly infectious to sand flies.

## 6. Systemic Immunity and Correlations to the Skin

The skin immune environment contributes to preventing inflammation and allowing the buildup of skin parasite patches. The immune environment transitions from an inflammatory to a regulatory environment as chronic infection is established, contributing to parasite accumulation in the skin (Figure 1). It has long been established that high levels of interleukin-10 (IL-10) in keratinocytes are a strong predictor of the development of post-kala-azar dermal leishmaniasis in patients treated for visceral leishmaniasis [58]. There is also a higher expression of programmed death-ligand 1 (PD-L1) and indoleamine 2,3-dioxygenase (IDO1) checkpoint molecules on CD68+ monocytes and macrophages infected with parasites compared with noninfected monocytes and macrophages [59].

After treatment, patients with *L. donovani* cutaneous leishmaniasis expressed reduced PD-L1 and IDO1 in macrophages [59]. An early reduction of PD-L1 expression predicted a faster rate of clinical cure in parallel with a reduction in parasite load [59]. Symptomatic *L. infantum*-infected dogs demonstrate a four-fold increase or two-fold increase in PD-1 surface expression on CD4+ T cells and CD8+ T cells, respectively, compared with healthy controls [38]. Blockage of PD-L1 with anti-PD-L1 antibody partially restored CD4+ and CD8+ T cell proliferation, CD4+ interferon-gamma (IFNy) production, and increased macrophage reactive oxygen species (ROS) production, altogether resulting in decreased macrophage parasite burden [60]. CD4+ T cell effector responses may also be important for preventing infectiousness of host animals; Guarga et al. found that *L. infantum*-infected dogs with lower CD4+ T cell counts had higher rates of infectiousness to the vector [61]. This supports a role for PD-L1 checkpoint inhibitor drugs used either topically or systemically as part of an antileishmanial regimen or PD-L1 expression as a predictor of clinical outcomes [59,60]. This also supports the idea of a regulatory environment in the skin being more conducive to parasite patch coalescence.

B cells are associated with PD-L1 and the creation of a regulatory environment as clinical VL progresses [60,62]. A cytosolic tryparedoxin from *L. infantum* specifically activated B cells and caused them to secrete IL-10 and immunoglobulin G (IgG) [63]. Regulatory IgD^hi^ B cells drive IL-10 production in progressive VL dogs from Brazil [62]. This illustrates the importance of IL-10 for establishing chronic, latent infection. Blockage of IL-10 receptor has shown success in achieving a sterile cure in chronically infected mice but did not reach the same level of efficacy in studies of human or dog cells [64]. Studies on PD-L1 and IL-10 receptor inhibitors together suggest potential therapeutic avenues to pursue a sterile cure in chronically infected individuals with VL. PD-L1, B cells, and IL-10 all contribute to a regulatory skin environment and the establishment of chronic dermal *Leishmania* infection.

Immune cell and cytokine profiles of the skin can show clear delineations between infected and uninfected dogs. Immunohistochemistry studies of VL skin have demonstrated more cells of all types, particularly CD8+ T cells and macrophages, and increased IFNy and inducible nitric oxide synthase (iNOS) production in the dermis of lesioned skin from patients with *L. infantum* nonulcerated cutaneous leishmaniasis compared with the skin of healthy individuals [65]. Experimental studies looking at immune gene expression in leishmanin skin test positive reactions of Ibizan hounds saw significant upregulation of Toll-like receptor 2 (TLR2), IL10, IFNy, and PDL1 compared with the healthy skin of endemic control hounds [66]. Toll-like receptor 7 (TLR7) was significantly downregulated [66]. A prevalence of CD3+ and large mononuclear cells, as opposed to B cells, coupled with a high expression of TLR2 indicate an ongoing delayed-type hypersensitivity response during these skin test reactions [66]. Upregulation of these TLR genes also aligned with a competent immune response in these dogs and support a role for TLR2 as protective against *Leishmania*. In lesional skin from *L. infantum*-infected dogs, there were increased neutrophils, histiocytes, T cells, and GATA3+, and IL-17a+ cells compared with the skin from uninfected dogs [67]. Normal skin from infected dogs also had more histiocytes, T cells, and GATA3+ compared with uninfected dogs but was also more abundant in FoxP3+ cells [67]. These findings support the role of dermal IFNy-mediated inflammation in protective immunity. It is interesting to note the upregulation of FoxP3+ cells in normal skin from uninfected dogs since studies in *L. infantum*-infected murine spleen and lymph nodes have shown high levels of CD4+ CD25+ regulatory T cells and FoxP3 expression that may contribute to immunosuppression of effector responses and subsequent control of immunopathology [68]. It is possible that a similar mechanism is responsible for immunosuppression in the skin and protection from lesions in chronically infected dogs and people. Lesions and ulcers can also become infected with bacteria, leading to inflammation that can be protective against *Leishmania* in the skin [69].

## 7. Sand Fly Saliva, Neutrophils, and Macrophages in Skin

As previously described, the presence of macrophages in the skin can contribute to both parasite accumulation and burden. Most of what is known about initial events after *Leishmania* infection is from cutaneous *L. major* mouse experimental infections. In this setting, there was rapid and sustained neutrophil infiltration at localized sand fly bites from which neutrophils captured *Leishmania* parasites [70]. Neutrophils were attracted to the wound by interleukin-1B (IL-1β), CXCL1, and yellow salivary proteins of the sand fly [71]. Sand flies egest part of their gut microbiome with every bite, which triggers inflammasome production of IL-1β, sustaining neutrophil recruitment and infiltration at the bite site [72]. Sand fly salivary proteins contribute to sustained immune cell recruitment and the establishment of initial infection and dermal parasite burden. Lymphocyte antigen 6 complex locus G6D (Ly6G+) neutrophils rapidly invade the dermis after parasite inoculation, followed by macrophages, natural killer (NK) cells, and then B cells [73]. In addition to sand fly microbiota and saliva components, infected sand fly bites also deliver parasite-produced components promastigote secretory gel (PSG) and *Leishmania* exosomes into the skin [74,75]. PSG is a powerful macrophage recruitment molecule in the skin and enhances the synthesis of polyamines necessary for intracellular parasite growth [75]. *Leishmania* exosomes are vesicles with a proinflammatory capacity and the ability to recruit neutrophils to the bite site, exacerbating the influx of innate immune cells that can be infected and harbor parasites [74]. Neutrophils create an epidermal block within 1 hour of a sand fly bite [70]. It might be expected that neutrophils aid in parasite clearance; however, depletion of neutrophils at the bite site actually reduces the number of viable parasites in the skin, one of many *Leishmania* immune evasion strategies [70]. Parasite burden then shifts to macrophage populations, without immune activation, via phagocytic uptake after 6–7 days, according to a model that has been dubbed the “Trojan Horse theory” [70]. The mannose receptor is part of the C-type lectin family with the ability to bind and internalize many different endogenous and pathogen-derived ligands for antigen processing and presentation [76]. In *Leishmania*-resistant C57BL/6 mice, *L. major* produces nonhealing cutaneous lesions where parasites can be taken up by mannose receptor (MR)-positive dermal macrophages exhibiting M2 characteristics despite systemic Th1 inflammatory states [77]. M2 macrophages are alternatively activated by exposure to cytokines such as IL-4, IL-10, or IL-13 and are associated with wound healing and tissue repair via proliferation induction [78]. This MR^high^ dermal macrophage population is maintained near the lesion by IL-4 and IL-10 [77]. Newer studies indicate that dermis-resident macrophages are the predominant phagocyte to engulf *L. major* in a murine model in the first 24 h of transmission by sand fly bite [79]. Confocal imaging shows infected neutrophils transferring their phagocytosed parasites to tissue-resident macrophages (TRMs) in situ, and TRMs engulf apoptotic-infected neutrophils in the skin in vitro [79]. RNA transcriptomic data from lesions in patients with CL demonstrated significantly higher expression of PD-L1 on parasite-infected monocytes and macrophages [59]. Less is known from VL patients, but a high number of M1 macrophages were detected in skin biopsies from patients infected with *L. infantum*, which also causes nonulcerated CL [80]. Cytosolic tryparedoxin secreted by *L. infantum* is one of many immune evasion pathways that helps the parasite combat ROS in macrophages, contributing to parasite resistance to the innate immune system [63]. Sand fly salivary proteins acted as chemoattractants leading to subsequent recruitment of macrophages for parasitism at the bite site, which allowed parasites to build up in the skin, completing the early stage of infection. During natural infection in dogs, *L. infantum* parasites produced increased RNA transcripts of metalloproteases in the skin, particularly at ear edges [81]. These proteases may serve as virulence factors contributing to dermal parasite burden and dermal tropism.

## 8. Sand Fly Bites and Skin

Events early in infection directly after the sand fly bite contribute to the dermal immune environment altering parasite persistence and transmissibility. When sand flies damage blood vessels, erythrocytes leak into tissue and are phagocytosed by macrophages, leading to the production of heme-oxygenase-1 (HO-1) [82]. HO-1 is also induced by sand fly saliva at bite sites early in the infection process in a mouse model [83]. Carbon monoxide is a major end product of HO-1 enzymatic reactions and suppresses inflammation in the skin and in infected cells, increasing parasite survival [83]. Inhibition of HO-1 enhances inflammation and subsequent tissue damage [82]. HO-1 from erythrocytes was induced by sand fly saliva to be an immune evasion tactic by *Leishmania*, as shown in a murine model. By suppressing inflammation in the skin, parasites thrived, contributing to parasite burden in mice.

Along with HO-1 induction, sand fly inflammatory salivary antigens elicited localized inflammation in murine models [57,71,72,84]. Sand fly salivary components are known to be vasodilating and immunomodulatory. Adenosine and AMP found in *Phlebotomus* species saliva had both properties [85]. Maxadilan from *Lutzomyia* sand fly species was a potent vasodilator [86]. Antimaxadilan vaccination protected mice from *L. major* infection [87]. Sand fly saliva has immunomodulatory effects on murine T cells, dendritic cells, and neutrophils [88]. Dogs immunized with 2 of 35 sand fly salivary proteins developed a strong cellular immune response at the bite site, characterized by increased IFNy and IL-12 production [84]. Although sand fly saliva is generally anti-inflammatory, resulting in less inflammation during parasite inoculation, prior immunization against parasites causes saliva-induced local inflammation at a nascent bite site, resulting in an adaptive immune response to saliva that can escalate to severe allergic responses and anaphylaxis. Sand fly saliva components are currently being considered for vaccine candidates because of their immunomodulatory properties. Similar to HO-1 enzyme, sand fly saliva suppresses inflammation in the skin, potentially allowing for parasites to evade immune destruction and thrive.

Keratinocytes are the primary epithelial cell in the skin and make up much of the dermal cell repertoire. Keratinocyte interplay with immune cells and damage to keratinocytes can set up the immune milieu occurring at the skin interface at the time of parasite exposure. Exposure of human keratinocytes to different species of *Leishmania* elicits distinct pro- or anti-inflammatory responses. The epidermis is a major source of cytokine expression that can determine the transition from a protective Th1 inflammatory response to a regulatory Th2 switch resulting in disease progression [89]. Mice resistant to *L. major* infection demonstrate significant gene production of IL-12, IL-1B, and IL-4 in the epidermis, promoting Th1 differentiation and resistance during the time of Th1/2 differentiation [89]. Supporting this, other experiments using murine keratinocytes incubated with *L. infantum* demonstrated upregulation of inflammatory cytokine genes for IL-6, IL-8, TNF-a, and IL-1b [90]. In a murine model, parasites accumulate in phagocytic cells in the reticular dermis more than in the epidermis or hypodermis [54]. These studies exemplify the role of keratinocytes in eliciting a proinflammatory response at the bite site and initiating disease. Keratinocyte contribution to a proinflammatory environment has an effect on parasite persistence in the skin and subsequent outward transmission of parasites.

## 9. Summary

VL is an important neglected tropical disease with a significant global burden. Understanding what effects transmission is crucial for control of the disease. VL is unequivocally dermotropic even in visceralizing species, and skin parasite burden best predicts transmission. Parasite-driven events and dermal immune mechanisms control parasite skin burden. Sand fly salivary antigens and HO-1 induction promote both inflammation and immunosuppression, as well as the recruitment of favored host cells—macrophages—to the bite site. Parasitism of recruited macrophages and propagation into parasite patches establishes parasite persistence in the skin. Expression of PD-L1 and IL-10 secreted by regulatory B cells contribute to the transition from a proinflammatory to regulatory environment systemically, which is likely to directly impact the establishment of chronic dermal infection. The immunology of the skin, in turn, influences skin parasite burden. Densely parasitized patches of the skin provide an ideal transmission situation for sand flies from both human patients and naturally infected canine reservoirs.

## 10. Conclusions

Further examination of the dermal immunology of the skin as it pertains to *Leishmania* infection is important for understanding the transmission of visceralizing *Leishmania* spp. Further elucidation of immunological mechanisms involved in outward transmission from the skin to sand flies will inform the creation of topical immunomodulatory therapeutics that can prevent outward transmission. Topical PD-L1 inhibitors or IL-1 receptor antagonists could decrease dermal inflammation of infected hosts and prevent vector transmission from infected dogs or people if specific cytokines are found to be associated with increased infectiousness. Establishment of *L. infantum* persistence in the skin and the dermal immune environment of humans infected with VL must be further explored. Understanding parasite trafficking to the skin and immune regulation of skin parasite burdens is critical for intervening in transmission and curbing the infection cycle for One Health.

## Figures and Tables

**Figure 1 pathogens-11-00610-f001:**
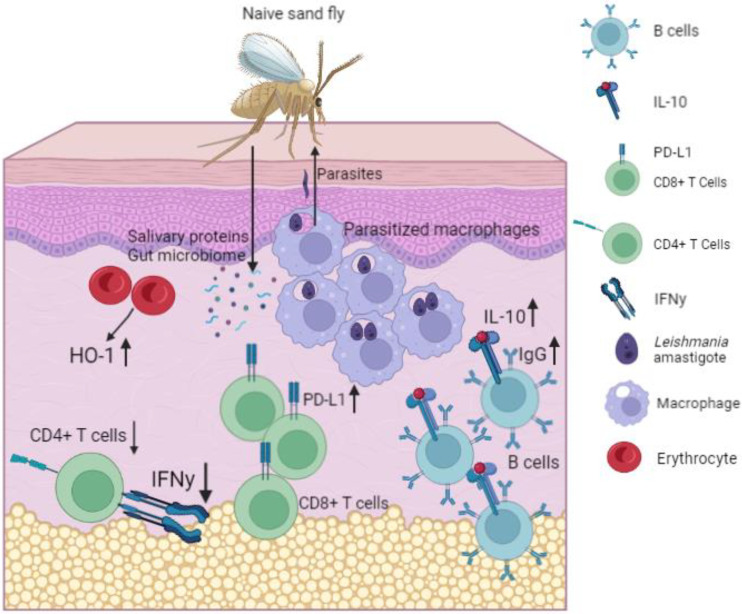
**Outward parasite transmission to naïve sand flies in a model of chronic *L. infantum* infection.** Parasitized macrophages cluster together in the skin in a self-propagating manner and heterogenous, patchy dermal distribution. Establishment of chronic infection in the skin is promoted by decreased CD4+ T cell effector responses, decreased IFNy secretion, increased PD-L1, hypergammaglobulinemia, and increased B-cell IL-10 secretion. When naïve sand flies feed, amastigote-parasitized macrophages are transmitted from the skin into the sand fly gut, infecting the sand fly. The bite site receives sand fly salivary proteins, sand fly gut microbiota, and an increase in erythrocyte heme-oxygenase-1 production, all of which contribute to an anti-inflammatory environment and subsequent parasite persistence. Created with BioRender.com, accessed on 10 March 2022.

## Data Availability

Not Applicable.

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
