# Peer review of "Visceral Leishmaniasis and the Skin: Dermal Parasite Transmission to Sand Flies"

_pathogens, 2022, doi:10.3390/pathogens11060610_

Round 1

Reviewer 1 Report

I reviewed the Manuscript (pathogens-1653933-peer-review-v1) entitled “Visceral Leishmaniasis and the Skin: Dermal Parasite Transmission to Sand Flies” by Arumugam and colleagues. This manuscript lacks an introduction including the rationale and aims of the review. Though limited information is reported in the poor abstract (it is not an objective and comprehensive representation of the article), the reason why this review is necessary and what implications (also absent from the conclusions) it might have completely lacked.

Furthermore, though this article is not a systematic literature review, it is highly advisable to explain how the review was carried out: what database was consulted? what search/reading keys were used? what type of articles were included/excluded from the review (e.g., original articles, case reports, conference proceedings, etc.)? It would be also appropriate to indicate the time frame considered for the research. The method used to construct a Review article is essential not only for the 'reproducibility of the research' but also to evaluate which evaluation criteria were used by the Authors to critically write the manuscript.

Author Response

Manuscript ID: pathogens-1653933

Type of Manuscript: Review

Title: Visceral Leishmaniasis and the Skin: Dermal Parasite Transmission to Sand Flies

Authors: Sahaana Arumugam, Breanna M. Scorza, Christine Petersen

Received: March 10 2022

We thank the academic editor and reviewers for their thorough review and insightful comments regarding our manuscript and welcome the suggestions for improvement. Below we list the comments from each of 4 reviewers and our response to each comment. We appreciate positive comments but have not included them below. Any changes made based on comments were included in our tracked changes of the marked copy. We have submitted a clean copy of the revised manuscript as well. 

Reviewer 1:

This reviewer indicated that the manuscript could be immensely improved. They stated that “This manuscript lacks an introduction including the rationale and aims of the review… the reason why this review is necessary and what implications it might have.” They also stated that the abstract was poor and was not an objective and comprehensive representation of the article. This reviewer also included suggestions for including methodology of how this review was conducted for reproducibility purposes such as “what database was consulted? What search/reading keys were used? What type of articles were included/excluded from the review….time frame…”

Response:

We appreciate the reviewer’s suggestions for improvement regarding fleshing out our abstract to make it more comprehensive and providing a thorough introduction detailing the aims and rationale for the review. We also appreciate the advisement regarding more detailed methodology for how articles were selected for this literature review. We have expanded the abstract to encompass the various sections in the article and the main ideas of each section to make sure that it stands on its own and is more comprehensive. We have also included an introduction to detail the goals of the review, why we think this review is necessary and what the implications of the big idea of the review are. This is not a systematic literature review; regardless we have taken the comments about literature review methodology into account and added a few sentences regarding the search parameters for the literature search such as the database, terms, types of articles and year range used.

Reviewer 2 Report

The review contributes to a better understanding of the role of the skin in the transmission of Visceral leishmaniasis. I have a few concerns below.

1)Line 23- According to WHO (2022) in 2020 more than 90% of cases occurred in 10 countries: Brazil, China, Ethiopia, Eritrea, India, Kenya, Somalia, South Sudan, Sudan, and Yemen. 

2) Line 34- when the authors say "Parasites from vertically infected North American hounds are still infectious to competent sand flies and thus pose a risk of outward domestic transmission to proximate human populations, particularly in the Southwest, South and Southeastern U.S. where sand flies are endemic." The thought is that may occur the possibility of transmission of L. infantum by sand flies in the USA. Psathyromyia shannoni is not vector according to the criteria of vectorial competence (Ready 2013). Therefore the authors should rewrite better this sentence.

3)Line 82-  The authors say "It is likely that L. infantum parasites persist in human skin and potentially promote transmission" but, unfortunately, there is no proof that humans can infect sand flies. There is only a discussion that individuals with HIV could transmit L. infantum to sand flies. I suggest the authors should rewrite better this sentence.

3) It is necessary to describe each immune abbreviation, at least for the first time (PD-L1, IDO1 for example).

4) From item 6 (line 227) forward, all scientific names are without italics.

Author Response

Manuscript ID: pathogens-1653933

Type of Manuscript: Review

Title: Visceral Leishmaniasis and the Skin: Dermal Parasite Transmission to Sand Flies

Authors: Sahaana Arumugam, Breanna M. Scorza, Christine Petersen

Received: March 10 2022

We thank the academic editor and reviewers for their thorough review and insightful comments regarding our manuscript and welcome the suggestions for improvement. Below we list the comments from each of 4 reviewers and our response to each comment. We appreciate positive comments but have not included them below. Any changes made based on comments were included in our tracked changes of the marked copy. We have submitted a clean copy of the revised manuscript as well. 

Reviewer 2 

This reviewer’s comments were generally very positive, which we greatly appreciated. They had several specific suggestions regarding changes for improvement and clarity which we very much appreciate. Below are our responses to their comments:

Reviewer: They indicated “According to WHO (2022) in 2020 more than 90% of cases occurred in 10 countries: Brazil, China, Ethiopia, Eritrea, India, Kenya, Somalia, South Sudan, Sudan and Yemen.”

Response: This line was altered to reflect the more recent WHO data as suggested by the reviewer using the WHO Leishmaniasis data sheets.

Reviewer: They indicated “when the authors say "Parasites from vertically infected North American hounds are still infectious to competent sand flies and thus pose a risk of outward domestic transmission to proximate human populations, particularly in the Southwest, South and Southeastern U.S. where sand flies are endemic." The thought is that may occur the possibility of transmission of L. infantum by sand flies in the USA. Psathyromyia shannoni is not vector according to the criteria of vectorial competence (Ready 2013). Therefore the authors should rewrite better this sentence.”

Response: We understand the reviewer’s point about the criteria of vectorial competence from Ready 2013. We have rephrased this language to better reflect that there are no proven vectors in North America for L. infantum. However we chose to cite several sources in our rephrasing (Schaut 2015, Koch 2017, Chalghaf 2018) which demonstrate that there have been autochthonous cases of L. mexicana infection within the contiguous United States vectored by L. anthrophora or L. diabolica (Schaut cites Texas, Kansas, Missouri and Oklahoma as having cases). We speculate that with climate change and other environmental factors potentially changing vector distribution, the existence of known vectors as far north as Mexico, and cases of autochthonous transmission in the US, there may be a potential future issue of zoonoses from naturally infected dog reservoirs of L. infantum in certain areas of the US. L. shannoni was shown to possibly be a vector for L. infantum in Brazil (Travi, 2002). Although this is not yet a confirmed problem we are taking a one health perspective and thinking about it as a potential emerging threat in areas with both infected dogs and L. shannoni in North America.

Reviewer: They indicated altering the sentences discussing "It is likely that L. infantum parasites persist in human skin and potentially promote transmission" but, unfortunately, there is no proof that humans can infect sand flies. There is only a discussion that individuals with HIV could transmit L. infantum to sand flies.”

Response: We understood the comment to refer specifically to the lack of proof regarding L. infantum parasite transmission from humans to vectors. We have changed this language to more accurately reflect the references we provided that show proof of L. donovani transmission from humans to vectors in xenodiagnoses studies. We have also included one reference (Molina et al, 2020), that demonstrated L. infantum transmission in a xenodiagnoses study from 2 immunocompetent patients with VL to sand fly vectors.   

Reviewer: The reviewer indicated that “It is necessary to describe each immune abbreviation at least for the first time” and that “from item 6 forward all scientific names are without italics”.

Response: We appreciate the reviewers comments regarding these stylistic inaccuracies. We have spelled out each immune abbreviation the first time it is in use. We have also changed all scientific names to italicized names.

Reviewer 3 Report

This is a good review of an emerging topic.

A few minor points:

Clarify what Hanson's disease is and its significance.

This reference is of particular importance, and should be incorporated into the review:

Sing et al 2021. Lancet Microbe. Xenodiagnosis to evaluate the infectiousness of humans to sand flies in an area endemic for visceral leishmaniasis in Bihar, India: transmission-dynamics study.

Author Response

Manuscript ID: pathogens-1653933

Type of Manuscript: Review

Title: Visceral Leishmaniasis and the Skin: Dermal Parasite Transmission to Sand Flies

Authors: Sahaana Arumugam, Breanna M. Scorza, Christine Petersen

Received: March 10 2022

We thank the academic editor and reviewers for their thorough review and insightful comments regarding our manuscript and welcome the suggestions for improvement. Below we list the comments from each of 4 reviewers and our response to each comment. We appreciate positive comments but have not included them below. Any changes made based on comments were included in our tracked changes of the marked copy. We have submitted a clean copy of the revised manuscript as well.

Reviewer 3

This reviewer has generally very positive comments, which we greatly appreciated. There were two specific minor points they had indicated. They asked us to “Clarify what Hanson’s disease is and its significance.” They also requested that we incorporate a specific paper (Singh, 2021, Lancet Microbe) into our review due to its particular importance.

Response: We appreciate the reviewer’s minor comments and suggestions for improvement. We have changed the sentences regarding Hansen’s disease to clarify what Hansen’s disease is and elaborate on why this is a significant comorbidity. Hansen’s disease causes significant dermal lesions and these breaks in the skin may increase susceptibility to parasite accumulation in the skin as well as change the dermal immune environment. Regarding the Singh et al article, this article has been cited and discussed at length in the original manuscript in the section titled “Transmission and Dermal Parasite Burden”.

Reviewer 4 Report

The review focuses on a highly relevant topic: factors affecting the distribution of Leishmania parasites in the host skin that are crucial for outward transmission. I highly appreciate the comprehensiveness of the review and the quality of the language and writing style and have only minor comments.

In Chapters 6 – 8, scientific names should be italicised.

Line 25: Leishmania infantum resides in canid reservoirs, whereas L. donovani is thought to be predominantly anthroponotic (the role of animal reservoir hosts has been suggested, but their identity has not been established).

Lines 59- 60: The passage describing the correlation between the severity of canine disease and infectiousness should be based on multiple references. The fact that clinical severity was inversely related to the infectiousness of dogs is also reported by Laurenti et al 2013 (http://dx.doi.org/10.1016/j.vetpar.2013.03.017) and Molina et al 1994 (https://doi.org/10.1016/0035-9203(94)90446-4) found no correlation. On the other hand, there are many studies showing a positive correlation between disease severity and infectiousness to sand flies: Travi et al 2001 (Am. J. Trop. Med. Hyg., 64), Courtenay et al 2002 (The Journal of Infectious Diseases 186), da Costa –Val et al 2007 (doi:10.1016/j.tvjl.2006.11.006), Michalsky et al 2007 (doi:10.1016/j.vetpar.2007.03.004), Vercosa et al 2008 (doi:10.1186/1746-6148-4-45), Soares et al 2011 (doi:10.1016/j.actatropica.2010.08.015),  MagalhÇŽes-Junior et al 2016 (http://dx.doi.org/10.1016/j.vetpar.2016.04.031). Importantly, a meta-analysis of data published up to 2009 by Quinnell and Courtenay (doi:10.1017/S0031182009991156) showed that the proportion of infectious dogs increases significantly with clinical severity.

Lines 82-83: Experiments describing the infectiousness of humans infected with L. infantum have also been performed and should also be cited (Molina et al 1999 Am. J. Trop. Med. Hyg. 60(1); Molina et al 1994 AIDS 8(2)). The phenomenon of the extremely high infectiousness of patients co-infected with HIV and VL should be mentioned.

Chapters 2 can be enriched also by the evidence that repeated multiple sand fly bites significantly increase L. major parasite loads in the skin (Vojtkova et al. 2021 doi: 10.3389/fitd.2021.745104).

The work of Guarga et al 2000 (https://doi.org/10.1053/rvsc.2000.0419) should also be cited, as these authors focused on the relationship between the immune status of infected dogs and their infectiousness and reported that the lower the CD4+T cell count, the higher the infection rate in the vector.

Chapter 6/7 and Figure 1: In addition to saliva and sand fly microbiota, sand fly bites also deliver  PSG and exosomes derived from leishmania into the wound, which enhance infection, are pro-inflammatory and affect macrophage functions (Atayde et al 20156, https://doi.org/10.1016/j.cellimm.2016.07.013; Rogers et al 2009, https://doi.org/10.1371/journal.ppat.1000555)

Lines 289-290: The sentence “sand fly saliva is anti-inflammatory in general to prevent inflammation during parasite inoculation” should be rephrased to not imply that sand fly saliva evolved to promote parasite development. 

Author Response

Manuscript ID: pathogens-1653933

Type of Manuscript: Review

Title: Visceral Leishmaniasis and the Skin: Dermal Parasite Transmission to Sand Flies

Authors: Sahaana Arumugam, Breanna M. Scorza, Christine Petersen

Received: March 10 2022

We thank the academic editor and reviewers for their thorough review and insightful comments regarding our manuscript and welcome the suggestions for improvement. Below we list the comments from each of 4 reviewers and our response to each comment. We appreciate positive comments but have not included them below. Any changes made based on comments were included in our tracked changes of the marked copy. We have submitted a clean copy of the revised manuscript as well.

Reviewer 4

This reviewer had extremely positive comments in general which we greatly appreciate. They had several minor suggestions regarding areas for expansion and references that we should include in our citation. Below is our responses to each comment.

Reviewer: The reviewer indicated that scientific names in chapters 6-8 should be italicized

Response: We greatly appreciated the reviewer catching this oversight for us and have corrected all scientific names to be italicized.

Reviewer: The reviewer indicated “Leishmania infantum resides in canid reservoirs, whereas L. donovani is thought to be predominantly anthroponotic (the role of animal reservoir hosts has been suggested, but their identity has not been established).”

Response: We appreciate the reviewers comment. We have changed the language to reflect L infantum as having confirmed canid reservoirs but L donovani being predominantly anthroponotic. We have also included two new citations supporting the suggestion of animal reservoir host existence for L. donovani but we chose not to elaborate further as considerable speculation on animal reservoirs is outside the scope of this article.

Reviewer: The reviewer indicated “The passage describing the correlation between the severity of canine disease and infectiousness should be based on multiple references. The fact that clinical severity was inversely related to the infectiousness of dogs is also reported by Laurenti et al 2013 and Molina et al 1994 found no correlation. On the other hand, there are many studies showing a positive correlation between disease severity and infectiousness to sand flies: Travi et al 2001, Courtenay et al 2002, da Costa –Val et al 2007), Michalsky et al 2007, Vercosa et al 2008, Soares et al 2011,  MagalhÇŽes-Junior et al 2016. Importantly, a meta-analysis of data published up to 2009 by Quinnell and Courtenay showed that the proportion of infectious dogs increases significantly with clinical severity.”

Response: We truly appreciate the reviewer’s thorough discussion about the points regarding severity of canine disease and infectiousness. We do agree that this point is an important one to flesh out and should have multiple references to support this theory. Several of the citations that the reviewer graciously provided were used to elaborate on clinical severity and the connection to infectiousness in the section titled ‘Transmission and Dermal Parasite Burden’.

Reviewer: The reviewer indicated ”Experiments describing the infectiousness of humans infected with L. infantum have also been performed and should also be cited (Molina et al 1999 Am. J. Trop. Med. Hyg. 60(1); Molina et al 1994 AIDS 8(2)). The phenomenon of the extremely high infectiousness of patients co-infected with HIV and VL should be mentioned.”

Response: We appreciate the reviewer pointing out a crucial phenomenon regarding outward transmission in co-infected HIV/VL patients. The experiments regarding co-infection and transmission in humans have been included.

Reviewer: The reviewer indicated that “Chapters 2 can be enriched also by the evidence that repeated multiple sand fly bites significantly increase L. major parasite loads in the skin”.

Response: We appreciate the citation the reviewer provided regarding this point from Vojtkova et al (2021) and we incorporated this point in conjunction with other xenodiagnoses experiments in section 3.

Reviewer: The reviewer indicated “The work of Guarga et al 2000 should also be cited, as these authors focused on the relationship between the immune status of infected dogs and their infectiousness and reported that the lower the CD4+T cell count, the higher the infection rate in the vector.”

Response: We appreciate this point the reviewer made and have included the citation with a brief discussion in the section detailing immune responses in the skin during VL.

Reviewer: The reviewer indicated “Chapter 6/7 and Figure 1: In addition to saliva and sand fly microbiota, sand fly bites also deliver  PSG and exosomes derived from leishmania into the wound, which enhance infection, are pro-inflammatory and affect macrophage functions

Response: We appreciate the reviewers comments regarding this and included the two citations the reviewer provided to cover egestion of PSG and Leishmania exosomes and their effect on bite site inflammation. However Figure 1 only outlines outward transmission from an infected host to a naive sand fly. Therefore we did not include the PSG and exosomes being inserted into the skin from the sand fly since the naive sand fly would not have any of these parasite products in the fly mid gut to egest into the bite site. 

Reviewer: The reviewer indicated “The sentence “sand fly saliva is anti-inflammatory in general to prevent inflammation during parasite inoculation” should be rephrased to not imply that sand fly saliva evolved to promote parasite development.”

Response: We appreciate the reviewer’s comments to improve clarity. We have amended the sentence so as not to imply specific evolution on the part of the sand fly saliva.